# Precision Oncology by Point-of-Care Therapeutic Drug Monitoring and Dosage Adjustment of Conventional Cytotoxic Chemotherapies: A Perspective

**DOI:** 10.3390/pharmaceutics15041283

**Published:** 2023-04-19

**Authors:** Myriam Briki, Pascal André, Yann Thoma, Nicolas Widmer, Anna D. Wagner, Laurent A. Decosterd, Thierry Buclin, Monia Guidi, Sandro Carrara

**Affiliations:** 1Service and Laboratory of Clinical Pharmacology, Department of Laboratory Medicine and Pathology, Lausanne University Hospital and University of Lausanne, 1011 Lausanne, Switzerland; myriam.briki@chuv.ch (M.B.); pascal.andre@chuv.ch (P.A.); nicolas.widmer@chuv.ch (N.W.); laurentarthur.decosterd@chuv.ch (L.A.D.); thierry.buclin@chuv.ch (T.B.); 2Bio/CMOS Interfaces Laboratory, École Polytechnique Fédérale de Lausanne—EPFL, 2002 Neuchâtel, Switzerland; sandro.carrara@epfl.ch; 3School of Engineering and Management Vaud, HES-SO University of Applied Sciences and Arts Western Switzerland, 1401 Yverdon-les-Bains, Switzerland; yann.thoma@heig-vd.ch; 4Pharmacy of the Eastern Vaud Hospitals, 1847 Rennaz, Switzerland; 5Institute of Pharmaceutical Sciences of Western Switzerland, University of Geneva, University of Lausanne, 1206 Geneva, Switzerland; 6Service of Medical Oncology, Department of Oncology, Lausanne University Hospital and University of Lausanne, 1011 Lausanne, Switzerland; dorothea.wagner@chuv.ch; 7Centre for Research and Innovation in Clinical Pharmaceutical Sciences, Lausanne University Hospital and University of Lausanne, 1011 Lausanne, Switzerland

**Keywords:** therapeutic drug monitoring, point-of-care, oncology, cytotoxics

## Abstract

Therapeutic drug monitoring (TDM) of conventional cytotoxic chemotherapies is strongly supported yet poorly implemented in daily practice in hospitals. Analytical methods for the quantification of cytotoxic drugs are instead widely presented in the scientific literature, while the use of these therapeutics is expected to keep going for longer. There are two main issues hindering the implementation of TDM: turnaround time, which is incompatible with the dosage profiles of these drugs, and exposure surrogate marker, namely total area under the curve (AUC). Therefore, this perspective article aims to define the adjustment needed from current to efficient TDM practice for cytotoxics, namely point-of-care (POC) TDM. For real-time dose adjustment, which is required for chemotherapies, such POC TDM is only achievable with analytical methods that match the sensitivity and selectivity of current methods, such as chromatography, as well as model-informed precision dosing platforms to assist the oncologist with dose fine-tuning based on quantification results and targeted intervals.

## 1. Introduction

Despite impressive advances in targeted anticancer therapies and immunotherapy, conventional chemotherapies are still fundamental constituents of treatment regimens for many cancer types. Unlike targeted treatments, which by definition have a wider therapeutic concentration range, conventional cytotoxic drugs are poorly specific cell poisons characterized by a narrow window of maneuver [1]. Insufficient exposure to the drug is commonly associated with a reduced treatment efficacy; on the other side, toxic adverse effects occur in the case of over-exposure. Although the dosage is typically adapted based on the patient’s body weight (BW) or body surface area (BSA), *a priori* precision dosing is far from being attained due to the widely acknowledged high inter-individual pharmacokinetic (PK) variability [2]. In such situations, therapeutic drug monitoring (TDM) would be expected to play a central role in cytotoxic treatment outcomes. TDM consists of measuring a drug concentration in a patient’s biological matrix, typically plasma, urine, or whole blood, and adjusting the drug dosage depending on the measured exposure to ensure a safe and efficient use of the drug [3]. In order to guide these adjustments, model-informed precision dosing (MIPD) can be particularly useful when available [4]. Although numerous authors have repeatedly advocated for its use [1,5,6,7], TDM of conventional chemotherapies is still rarely performed during the clinical management of oncological patients in real-life settings. High-dose methotrexate, busulfan, and 5-fluorouracil are the only cytotoxic agents for which TDM is recognized as beneficial based on compelling evidence [7].

Well established in disease areas such as immunosuppressants [8], TDM exploits the fundamental pharmacokinetic–pharmacodynamic (PK–PD) relationship linking the therapeutic and/or toxic effects of a drug with its circulating concentration (typically measured in plasma), which is known to be more precise than linking it with its prescribed posology [9]. In order to determine the concentration of a drug in a biological matrix, laboratories typically rely on highly sensitive and selective analytical methods such as high-performance liquid/gas chromatography coupled to mass spectrometry and immunoassays, which represent the current standards in routine TDM practice. Sample extraction procedures continue to become faster and more straightforward, allowing for the reduction of delays between sample collection and result transmission. Nonetheless, if access to a laboratory that operates continuously is limited, multiple issues can arise. Indeed, depending on the drug dosage regimen, delays occurring in traditional TDM may prevent the treatment from reaching its full potential. This is indeed the main issue with antineoplastic chemotherapies as they typically have timely intervals between the doses and/or are only administered over a short period of time, usually by intravenous infusion. This means that if a given dosage does not result in concentrations within the targeted therapeutic window, the opportunity for adjustment is then delayed to the next infusion, if any. Moreover, optimal adjustment could potentially need more than one cycle, leading to inefficient use of TDM [10]. Current TDM practices are mainly suited for *a posteriori* dose adjustment, which is performed sometimes routinely but more often only in cases of non-response or suspected toxicity to a treatment. This approach is satisfactory for several applications such as anti-epileptic therapy [11] or anti-infective therapies [12,13] when drugs are taken regularly and over a substantial duration. However, the usual setting of conventional chemotherapies is different, thus definitely limiting the feasibility of classical TDM. For these drugs, *a priori* precision dosing is challenging and constrained to rather elementary predictors such as BW, BSA, sex, age, markers of kidney or liver function, or even pharmacogenetic markers. Still other approaches including TDM after a low test dose have been investigated [14]. Between those two approaches lies *real-time dosage adjustment*, a perspective currently unavailable for chemotherapies yet not unknown to the medical community: it is widely applied in the case of glycaemia measures, for example [15]. To provide the opportunity for real-time dosage readjustment intervention to occur already during a drug infusion, a point-of-care (POC) device with built-in measurement, interpretation, and dosage recommendation tools would be optimal, possibly representing the next step in the history of TDM development [16]. Such a short-loop system should perform no worse than current standard analytical procedures, while allowing for real-time on-site quantification of the targeted analyte and computer-assisted decision support, so that it could be used by a wide range of healthcare professionals and not be restrained to qualified laboratory technicians and pharmacologists [17]. 

This perspective article aims to review the current limitations of TDM affecting the individualization of chemotherapies and the challenges that would meet an innovative POC-TDM system that integrates computer-assisted decision support. With this intent, six conventional antineoplastic drugs were selected. To provide a diversified overview, these particular examples included two cytotoxic compounds for which TDM is already largely available, two for which it is seldom offered, and two others for which no TDM exists: methotrexate (MTX) and busulfan (BSF), etoposide (ETP) and 5-fluorouracil (5-FU), and cyclophosphamide (CPA) and ifosfamide (IFS), respectively.

## 2. Examples of TDM for Cytotoxic Compounds

### 2.1. The First Cytotoxic Agent to Be Followed with TDM: Methotrexate

The oldest example of cytostatic chemotherapy subject to TDM is MTX, whose plasma concentration measurement was introduced almost 50 years ago [18]. Administered either orally or parenterally, MTX competes with reduced folates for active transport into cells by the reduced folate carrier 1 (RFC1), while at MTX serum concentrations exceeding 100 µmol/L, passive diffusion becomes dominant, allowing the drug to reach efficient intracellular concentrations [19,20]. MTX is then polyglutamated into its active metabolites through the action of folylpolyglutamyl synthetase and de-glutamated MTX (i.e., MTX-monoglutamate) is transported out of the cells for further excretion [6,20]. MTX and its metabolites are excreted renally. The parent compound also undergoes hepatic metabolism and is hydroxylated by the aldehyde oxidase into 7-hydroxy-MTX [20].

Used in high-dose treatments in oncology (Appendix A), MTX can cause a range of toxic effects, which include acute kidney injury, that influence its clearance (CL_MTX_) and therefore cause sustained exposure to the compound [21]. CL_MTX_ is also subject to high inter-individual variability, further affected by pharmacological interactions, particularly those with substances inhibiting the active renal tubular secretion of organic cations including MTX and its derivatives. Drug interactions reported to decrease CL_MTX_ and therefore elicit over-exposure to the drug include non-steroidal anti-inflammatory drugs (NSAIDs) (e.g., ketoprofen), antibiotics (e.g., piperacillin, amoxicillin, ciprofloxacin, and vancomycin), proton pump inhibitors (e.g., omeprazole and lansoprazole) [22], and sedatives and hypnotics (e.g., chloral hydrate) [23].

TDM is therefore recognized to be essential to ensure the safe use of this agent at high doses and, if necessary, to properly prescribe leucovorin rescue [6,24]. In accordance with dosage and pathology, target therapeutic ranges have been reported in the literature. A mean peak serum MTX concentration of between 1000 and 1500 μmol/L after a 6-h infusion of high-dose MTX is considered to ensure optimal antitumoral efficacy in the context of osteosarcoma [6]. For pediatric patients with acute lymphoblastic leukemia (ALL), a target range of 16–40 μmol/L at steady state (C_ss_, end of 24-h infusion) is advised [25].

TDM of MTX is routinely performed and highly advised in the context of high-dose regimens as this practice has led to a decrease in MTX-related severe toxicities [26]. Still, the objective is mainly to adjust the leucovorin rescue dosage and duration rather than to modify the MTX dose. The improvement of current TDM practices through bedside MTX measurement and dose modification could, however, greatly benefit oncology patients and refine the use of high dose regimens [27].

### 2.2. Consistently Subject to TDM: Busulfan

Commonly used as part of the conditioning regimen prior to hematopoietic stem cells transplantation (HSCT) [28], BSF-based treatments represent a preferred alternative to total body irradiation (TBI), especially in pediatric patients [29]. Typically, BSF is administered over 4 days in the form of 2-h infusions every 6 h (Appendix A) [28].

Both thoroughly and recently reviewed [30], the PK of BSF will only be briefly mentioned here to support further statements. Administered parenterally, unbound BSF diffuses into cells where it undergoes hydrolytic and enzymatic reactions to exert cytotoxic activity through its metabolites, namely tetrahydrofuran, methane sulfonate, and γ-glutamyldehydroalanylglycine [30]. Another metabolite, tetrahydrothiophene (THT), is transported out of the cells and undergoes hepatic metabolism to finally be excreted in the urine (BSF is excreted mostly in its metabolized form) [30]. Patient exposure to sub-therapeutic concentrations can lead to unfavorable outcomes such as disease relapse or graft failure while higher concentrations of BSF may result in acute toxicity and transplantation-related death [31]. Aside from its narrow therapeutic range, high inter-individual variability in the BSF PK profile has been observed [28]. Therefore, in view of the considerable number of off-target detrimental consequences and the narrow therapeutic window of BSF, TDM is widely considered as essential to ensure sufficient and safe BSF exposure, especially as the relationship between exposure and therapeutic outcome is established [31,32,33]. While multiple TDM approaches to assess drug exposure are available, the determination of the cumulative area under the curve (AUC_cum_) over the entire treatment period is the most suitable method for BSF exposure, with an optimum AUC_cum_ window of 78–101 mg∙h/L over the 16 doses of the 4-day treatment for HSCT conditioning [31]. Different sampling and computation strategies can be used to predict the individual AUC, but the most recommended one is relying on population pharmacokinetic (popPK) methods rather than traditional non-compartmental calculations [31,34]. However, actual practices still differ between TDM centers. Additionally, a two-compartmental model has been demonstrated to improve the accuracy of AUC calculations towards dosage adjustments [35].

Alternate methods such as the assessment of a low test dose right before high-dose treatment initiation combined with TDM were tested in an attempt to overcome the challenge of inter-individual variability [14]. The conclusion was that although this method provides a better estimation of the required dosage than weight-based dosing, the intra-individual variability still calls for intense adjacent TDM [14]. Indeed, patients undergoing BSF therapy typically experience a significant decrease in drug clearance (CL_BSF_) shortly after treatment initiation [28], which complicates the execution of TDM [36]. Such intra-individual variability means that TDM should ideally be performed at several administration points from treatment initiation to completion. A pharmacometabolomic approach has revealed correlations between ionic profile and CL_BSF_ and further highlights the potential interest of using additional tools alongside traditional TDM in order to optimize BSF therapy [37].

Nonetheless, TDM of BSF is largely applied in Europe as well as in the USA because of its well-recognized impact [31]. However, interventions based on TDM results still mostly rely on restricted sampling and methods with non-negligible turnaround times, which currently limits the optimization of BSF TDM.

### 2.3. Sporadically Subject to TDM: Etoposide

The topoisomerase II inhibitor ETP is used in the treatment of solid tumors and hematological malignancies and is particularly effective in cells presenting higher topoisomerase II levels [38]. 

ETP is commercialized as oral and intravenous preparations, the latter available as both native ETP and a phosphate-bound prodrug. The absolute bioavailability of oral etoposide is around 50% [39]. Concerning the injectable forms of ETP, they are considered equivalent from a pharmacokinetics point of view at the frequent 100 mg/m^2^ dose, even though phosphate-ETP shows a nearly 10% higher AUC [40]. Typically, intervals of multiple weeks are used between treatment cycles of ETP (Appendix A).

In addition, ETP pharmacokinetics is prone to significant inter- and intra-individual variability due to its mixed renal and biliary excretion combined with a hepatic metabolism [41]. Moreover, ETP is highly bound to protein, and in this context, TDM was shown to effectively decrease patient inter-individual variability [42]. Dose-limiting adverse effects are mainly myelosupression with neutropenia and other hematological events [43]. 

There is no clear AUC target established yet. However, longer survival was associated with a higher AUC for patients with non-small cell lung cancer [44]. An expected AUC_0–24h_ range of 76.7–136.7 mg∙h/L (4.6–8.2 mg∙min/mL) for 100 mg/m^2^ has been reported for injectable ETP in children [42]. Similar expected AUC_0–24h_ values were found in adults, and having a little higher AUC for cycle 3 was correlated with a higher neutropenia grade in high-dose ETP for advanced germ cell tumors [45].

### 2.4. A French Specialty: TDM of 5-Fluorouracil

The second example of a cytotoxic agent that is monitored sporadically, more often in France than elsewhere, is the antimetabolite 5-FU, whose TDM was shown to improve treatment outcomes some 35 years ago [46]. Its effect is attained through the inhibition of thymidylate synthase by its active moiety fluorodeoxyuridine monophosphate (FdUMP) along with 5,10-methylene tetrahydrofolate and by the respective incorporation of the 5-FU metabolites fluoro-deoxyuridine triphosphate (FdUTP) and fluorouridine triphosphate (FUTP) into DNA and RNA [47,48].

TDM of 5-FU is implemented in the case of gastrointestinal cancers, for instance (as well as other malignancies such as those presented in Appendix A), yet remains sub-optimal as the dosage is often reduced too drastically or even aborted in the case of toxicity, leading to unmet therapeutic needs and treatment failure [47]. 5-FU TDM is recognized to be important for adjusting dosing strategies for patients presenting with dihydropyrimidine dehydrogenase (DPD) deficiency as the enzyme is responsible for the majority of 5-FU degradation to its inactive metabolites [49] and such a deficiency is typically preventively screened using genotyping and phenotyping methods in order to prevent major toxicities form occurring at treatment initiation [50]. 

The toxicity arising from over-exposure to the drug manifests as diarrhea, mucositis, fever, stomatitis, nausea, and myelosupression [51,52], and solid evidence suggests a higher level of toxicity of 5-FU-based combination chemotherapies in women [53,54]. In severe rare cases, acute leukoencephalopathy can appear 2 to 4 days after 5-FU treatment initiation and prompt its interruption [52], as this drug-induced adverse effect is typically dose-dependent [55,56]. Unsurprisingly, as seen with other chemotherapeutics, the PK profile of 5-FU displays high inter-individual variability, and it has been shown that up to 80% of patients receiving the drug are outside the optimal therapeutic range [57]. The link between exposure, in this case the AUC, and treatment outcomes of 5-FU underlines the value of TDM [58]. The reported target AUC over one dose (1600–3600 mg/m^2^) for 5-FU is 20–30 mg∙h/L in the context of the FOLFOX6 regimen in the treatment of colorectal cancer [59], for example, and although TDM exists and is shown to improve the therapeutic outcomes of 5-FU treatments [60], it is scarcely implemented due to its long turnaround time, rendering it unsuitable for 5-FU [57].

### 2.5. Not Currently Benefiting from TDM: Cyclophosphamide and Ifosfamide

CPA is widely used in the treatment of lymphomas as well as other malignant diseases and can be administered orally or by intravenous infusions [61] at conventional and high-dose regimens (Appendix A). It exerts its cytotoxic action after metabolic activation by cytochrome P450 (CYP) enzymes, namely by its hydroxylation, which is mediated mostly by the CYP2B6 and CYP3A4 isoforms, leading to 4-hydroxy-CPA in tautomeric equilibrium with aldophosphamide, the latter generating the cytotoxic phosphoramide mustard as a result of its spontaneous scission as well as the urotoxic compound acrolein by β-elimination in this same step [61]. Its CYP-mediated metabolism implies a large panel of drug–drug interactions (DDI) that may affect the PK profile of CPA, notably with CYP3A4 inhibitors such as azole antifungals, which are commonly prescribed to patients undergoing chemotherapy [62,63].

Of note, conditioning regimens prior to bone marrow transplant historically included CPA associated with TBI or BSF, yet these myeloablative regimens present major toxicities [64] that can be avoided by using fludarabine instead of CPA [65]. The use of CPA can indeed lead to multiple toxic adverse events such as hepatic veno-occlusive disease (VOD), for which the AUC for 4-hydroxy-CPA during the first course of treatment was found to be a significant predictor [66]. TDM could therefore allow for safer use of CPA, however to our best knowledge, TDM has not been implemented up to this date.

However, there exist multiple published methods [67,68,69] for the quantification of CPA and its 4-hydroxy-CPA intermediate activation metabolite within generally observed circulating concentrations that would allow for the implementation of TDM. Moreover, CPA displays a high inter-individual PK variability with high CL_CPA_ variation not explained by common covariates such as age, body weight, or dosing regimen in infants, for example [70]. The AUC of CPA displays a coefficient of variation (CV) of 62%, although normalized by the dose, in young children [70]. For TDM, the monitoring of the CPA hydroxylated metabolite as well as the carboxyethylphosphoramide mustard metabolite may be more clinically relevant than the measurement of CPA alone [71]. Reported therapeutic targets for 4-hydroxy-CPA and phosphoramide mustard are AUC > 50 µmol/L and AUC 325 ± 25 µmol/L, respectively, in the context of HSCT using a CPA-TBI conditioning regimen [71]. As previously stated, this treatment plan is no longer favored, and in general, PK–PD targets may vary depending on the pathology and treatment regimen. Nonetheless, TDM should be performed not only to guide dosage adjustment towards treatment efficacy, as clinical studies may unveil new target ranges, but mostly to reduce the occurrence of toxic adverse events.

Used in the treatment of a variety of cancers, the prodrug IFS undergoes auto-inducible enzymatic activation, which increases its CL_IFS_ [72]. The active metabolite, 4-hydroxy-IFS, is produced alongside another product, chloroacetaldehyde. The latter accounts for toxic effects on the central nervous system and is proven to cause intracellular glutathione depletion, which in turn may allow for 4-hydroxy-IFS to be more active in target cancerous cells—a mechanism that could explain the lack of resistance towards ifosfamide treatments compared to other alkylating agents [73].

The IFS PK profile displays high inter-individual variability with, for instance, CL_IFS_ variability in pediatric patients reported to be 43% [74]. Dose-related adverse events are of a wide range as they include gastrointestinal, dermatologic, nervous, hematologic, renal, endocrine, cardiac, and hepatic side effects [75]. TDM could be performed in the first cycle of treatment to guide the dose adjustment for the following cycles [75], yet this practice is not implemented despite the availability of sensitive quantification methods [76,77]. To this date, no reported therapeutic target range has been identified. However, popPK studies allow deviations from typically observed concentrations within a population receiving the drug at a set dosage to be determined. Although less relevant than a properly determined target range when performing TDM, this information can already serve as an indicator to situate the patient’s exposure to the drug within the usual levels observed in patients.

## 3. Unmet Needs in TDM for Chemotherapeutics

Applied or not, TDM has been actually repeatedly advised for this class of therapeutics [1,7], and analytical methods presently available allow for the quantification of a wide range of cytotoxic substances [78,79]. Additionally, TDM was shown to be cost-effective in oncology for both cytotoxic agents and targeted therapeutics in a recent review [80]. The question then remains: why is TDM not widely implemented for cytotoxics? The previously mentioned hindering factors can be classified into two main groups of issues, namely those related to usual drug dosing regimens and those related to quantification of exposure. Regarding dosing regimens, as summarized in Appendix A, cytotoxics are usually administered either as a single dose or over a short course of a few days (e.g., four days for BSF), and lengthy intervals are used between doses to allow the patient to recover from one cycle of chemotherapy to the next (e.g., up to a three-week interval for high-dose MTX). This is poorly compatible with the current TDM turnaround time if the aim is dose adjustment: the loop comprising sample collection, reception at the laboratory, analysis, biomedical interpretation, and finally transmission of the results and recommendations to the oncologist, is too lengthy [81]. As a result of the short treatment duration, the time to modify the dosage to achieve the desired exposure during the same cycle is too short. Moreover, one adjustment could be insufficient. Certain TDM protocols call for performing measurements during the first cycle and applying the resulting dosage adjustment to the subsequent cycles. However, intra-individual variability that may result from physiological changes over long intervals jeopardizes the accuracy of such delayed adjustment, so that TDM partly loses its potential [82]. 

Regarding the quantification of exposure, while defining a target concentration range is typically convenient for TDM, cytotoxics mainly have target AUC intervals [6] that can span over the whole duration of a treatment cycle, as seen with BSF [31]. Inconveniently, sparse samples do not allow accurate AUC predictions through simple non-compartmental calculations (i.e., not computer-assisted), making rich sampling required for better estimation [6]. Applying a Bayesian approach supported by a pre-established popPK model certainly improves AUC estimation [31] based on a reduced number of samples. Still, using BSF as an example, even with rich sampling over the first 6-h infusion of the 4-day treatment, the steady state AUC_ss_ fails to be accurately predicted due to progressive drift of drug clearance [28], which demonstrates that intensive TDM remains preferable over the entire treatment course [82]. Practicing TDM over the entire course of a short-duration treatment cycle would essentially be relevant when real-time adjustment is feasible, which is currently not the case. 

Considering these complications, TDM with a short turnaround time would definitely allow for a more systematic implementation of the practice while simultaneously improving the adequacy of drug concentration exposure and consequently, treatment outcomes [81]. Additionally, the more TDM is practiced and accessible, the more PK–PD data will become available to refine the optimum PK targets for each drug, thus allowing further optimization of therapeutic adjustment [17,83]. This makes POC TDM an ideal solution to overcome the hurdles of chemotherapeutics TDM, as it would allow on-site quantification of a sufficient number of samples in real-time, seamless dosage adjustment based on Bayesian model-informed interpretation, higher chances of reaching the optimal target exposure, and an opportunity to collect data for further refinements of therapeutic individualization.

## 4. Adapting TDM to Cytotoxic Agents Using a POC Approach

### 4.1. Point-of-Care Tests in Oncology Treatments

Still limited until a few years ago [84], the use of POC technologies in oncology has drastically evolved in recent years in both diagnostic and treatment practices [85,86], yet this trend does not involve TDM for the moment. The need for remote, inexpensive, and reliable POC TDM systems has been highlighted, notably in the instance of imatinib [87]. In this particular case, however, unwieldy and expensive analytical devices such as the gold standard LC-MS/MS remain non-replaceable by current spectroscopy or electrochemical sensing methods, mostly due to their lack of selectivity and their current inability to accommodate multiplexed analysis [87].

### 4.2. Current Sensors for Chemotherapeutics

POC devices may be either external or wearable. They definitely require strong-enough sensitive and selective detection methods, scaled down to also be reliable at the patient bedside in either a hospital setting or an outpatient clinic [16]. Promising biosensing technologies allow the detection of target compounds on remote devices with minimal sample preparation. For BSF, to our best knowledge, no biosensor has been developed yet, but feasibility is not excluded thanks to largely documented biosensing technologies for other conventional chemotherapeutic agents. In the case of MTX, for example, bioluminescent sensors (luciferase-based indicators of drugs—LUCIDS) are capable of quantifying the MTX serum concentration within clinically observed concentration ranges in small blood volumes collected onto paper [88], and some authors have devised methods based on chemically induced dimerization systems to quantify serum MTX [89]. CPA concentration is also reportedly measurable using biosensors, more precisely using electrochemically mediated sensing relying on its metabolizing enzyme CYP3A4, which provides clinically relevant sensitivity and selectivity [90]. Detecting 5-FU is achievable as well using molecularly imprinted polymers with the principle of RNA-type nucleobase pairing [91]. Multi-panel biosensors for ETP, IFS, and CPA using amperometric quantification based on electrochemical detection also show great promise [92].

### 4.3. Continuous TDM and Wearable Technologies

Aside from cytotoxic agents, continuous TDM monitoring is feasible as demonstrated by the minimally invasive wearable sensor tested on human volunteers for the real-time monitoring of phenoxymethylpenicillin in extracellular fluid [93]. Such investigations prompt promising advances: the device does not disrupt skin integrity at the application site, therefore ensuring the patient’s comfort, and quantifies the drug concentration directly in vivo with up to 200 readings per second [93]. Although still in need of improvements, the principle demonstrates that wearable sensors can be developed without causing discomfort to the patient. Wearable and implantable drug monitoring devices and prototypes have recently been thoroughly reviewed [94]. The perspective is particularly relevant in the view of continuous TDM, which offers the most accurate description of the entire PK profile of the drug throughout the duration of a treatment cycle. Additionally, such data would be of great use to elaborate informative drug models, as rich sampling during popPK studies are scarce and labor-intensive. The principle of continuous TDM has been explored in the case of immunosuppressants: a quasi-continuous approach using micro-dialysis and optical immunosensing was compared to the traditional analysis of whole blood samples with LC-MS/MS, leading to the conclusion that this approach is not yet optimal in terms of analyte recovery and extraction efficiency, which are often problematic in intravenous micro-dialysis approaches [95]. Nevertheless, improvement of this approach is encouraging in terms of real-time continuous TDM and would in particular be greatly beneficial to the TDM of cytotoxic agents.

## 5. Model-Based Dose Adjustment as the Ultimate Goal of POC TDM

### 5.1. Reliability of the Interpretation and Recommendation Platform

Towards autonomous POC TDM systems, the interpretation of the results and the subsequent recommendations for dosage adjustment represent critical aspects [16]. Without strong safety features, the device cannot be considered as fully remote and autonomous. While the traditionally slow turnaround time of TDM allows for the intervention of a clinical pharmacologist or pharmacist to supervise the clinical utilization of concentration measurements, the information provided by a short-loop TDM device must be comprehensive and of utmost accuracy [17]. Such a precise method of interpretation will be made possible based on two main components: a powerful PK-predictive model and smart Bayesian adaptation algorithms. PopPK is the foundation of Bayesian or model-informed precision dosing: based on population data and relevant individual characteristics of the patient, a drug concentration measured in a patient gives rise to predictions of the subsequent individual PK curve, taking into consideration the applied dosage [96]. If the patient’s predicted PK curve does not fall within the desired range of suitable exposure, a dosage readjustment can be made based on model predictions so that the target exposure is achieved. This contrasts with traditional TDM, where if a measured concentration does not fall within the target range, a dosage adjustment is made by a simple proportionality rule, for example [97], which requires, first, the concentration profile to be at steady state and, second, sampling to be performed at a certain single and specific time, typically at trough [8]. In addition, infrequent TDM does not take into account variations in the individual PK profile over the treatment course [97]. Thus, *a posteriori* dose adjustment using MIPD is preferable, especially for drugs such as cytotoxics that are administered over short cycles separated by long intervals and display high inter- and intra-individual variability. 

When several popPK studies have been conducted for a given drug, datasets upon which models are built can vary substantially between studies, and the elaborated models will consequently differ [98,99]. The choice of the popPK model to implement in a POC TDM system is crucial in order to ensure correct interpretation and forecasting, as model-based predictions will directly reflect the quality of the data. Poorly designed studies may lead to biased or uninformative models. Another point to consider is the adequacy of the model in relation to the clinical setting: as a trivial example, a popPK model developed on adult cohort data will not be adapted to predict drug behavior in pediatric patients. Similarly, inpatients might significantly differ from outpatients. In addition, some influential factors that impact drug disposition may not have been explored within a given popPK analysis. For example, a certain co-medication may not be typically used in the center where the study patients were treated, yet it could be routinely part of the treatment elsewhere and be revealed to be a significant covariate modulating drug variability. The acceptability of a model relies on both its accuracy and the appropriateness of its intended use [96]. In certain situations, a metamodel based on individual and/or aggregated study data might better fit practical use than any of the published models available, as exemplified with tacrolimus [100]. 

In order to provide oncologists with assistance in dosage adjustment without relying on the presence of a pharmacologist or pharmacist, model-informed precision dosing platforms already exist and allow for the implementation of developed popPK models to guide clinicians [101]. A recent example is the TUCUXI software (http://www.tucuxi.ch/ (accessed on 15 February 2023)), aimed at both pharmacologists and non-professionals for guidance in dosage adjustment [102]. Other similar softwares include BestDose (https://bestdose.software.informer.com/ (accessed on 15 February 2023)), InsightRx (https://www.insight-rx.com/ (accessed on 15 February 2023)), DoseMe (https://doseme-rx.com/ (accessed on 15 February 2023)), and TDMx (http://www.tdmx.eu/ (accessed on 15 February 2023)) [96]. The implementation of such a platform into a POC TDM system is crucially needed for the real-time adjustment of dosages based on model predictions and targeted exposure.

### 5.2. Precision Dosing: Convergence between PK and PD Data

Providing that exposure–response relationships are convincingly known, model-informed precision dosing with Bayesian forecasting represents the best approach to ensure optimal dosage adjustment [103]. The combination of PK and PD data indeed offers insights into the probability of therapeutic efficacy and adverse events. For example, the risk of neutropenia is a major limiting factor in oncology when using chemotherapies, and Bayesian data assimilation using PK–PD models can provide a quantitative risk for each neutropenia grade [104]. This could allow mitigating decisions for dose increases to be made and enable rational treatment optimization. While several popPK studies can be found for most chemotherapeutic agents used in oncology, a comprehensive characterization of exposure–response relationships is, however, more difficult to find and traditionally represents a forgotten part of pharmacometrics in drug development [105]. Model-based development has markedly progressed in recent years and receives increasing attention from registration authorities and consequently, drug manufacturers. Additionally, methodological evolutions such as reinforcement learning incorporated into popPK modelling are expected to improve model performance on both the predictive side and in terms of computational resources [106] by using smaller, context-fitting sets of patients rather than large cohorts [107]. Such approaches could also be called upon to play some role in TDM. 

Using a PK–PD approach in TDM has also been called target concentration intervention (TCI). This approach relies on precise PK–PD interpolation to target specific concentrations for a specific desired therapeutic outcome. TDM traditionally aims at keeping the concentration exposure in a predefined range presumably associated with good efficacy and safety while TCI instead aims at optimally targeting therapeutic PD-guided concentrations [108]. During the development of chemotherapy doses and regimens, the maximum tolerated dose (MTD) is typically used as a reference, with the aim of minimizing the risk of non-response to the treatment and drug-resistance. However, this approach is unable to identify optimal doses, especially for subgroups of patients and different indications [109]. Translating this concept into a maximum tolerated concentration exposure requires the support of PK–PD approaches in order to define a target concentration or range. PK–PD modelling benefits from the full characterization of the disease and its response to treatment, reflected in markers such as the tumor size and type as well as the oncologic stage, together with their relationship with drug concentrations [110]. These data can most often not be obtained in early clinical trial phases as only a small number of patients receive the drug, making the MTD a more easily exploitable metric. Nonetheless, the increasing popularity of PK–PD modelling allows for a better apprehension of the dose–response relationship in newly developed drugs [111], such as new generation targeted therapeutics, as an efficacy plateau is expected due to their targeted properties [112]. The implementation of PK–PD approaches should, in the future, permit TDM and TCI approaches to be generalized. Although the rationale for not distancing from the MTD is clear, optimizing doses of historic cytotoxics at the level of the individual patient is also pertinent, precisely because dosing close to the MTD entails significant risks and amounts of toxicity to the patient. Thus, TDM of this class of drugs might have a higher benefit than it does in many other therapeutic classes where it is more widely implemented. 

### 5.3. TDM as a Source of Continuous Learning

A more widespread use of approaches to monitor drug concentrations and adjust dosages will enable the large-scale acquisition of corresponding data associated with therapeutic response and safety markers [83]. Such databases consolidating measurements from a large number of cancer units in an automated fashion will allow for fruitful in-depth research into the PK–PD characteristics of drugs in the target population of patients receiving them. This wealth of data will increase knowledge and progressively refine TDM and TCI strategies to optimize the therapeutic value for patients [17]. Thus, on top of the “short loop” offered to patients by POC TDM, a “long loop” of feedback will develop, refining the effectiveness and relevance of TDM itself.

## 6. Conclusions

This perspective paper aimed at exposing the current limitations of the traditional TDM procedures and practices regarding the administration of cytotoxic chemotherapies. At the same time, this paper offers insight into POC TDM and related features to succeed in the efficient and meaningful monitoring of drug levels. Current long turnaround times are poorly compatible with the dosing schemes applied for these drugs, yet theoretical criteria indicate that TDM would be highly suitable to improve treatment efficacy and safety. The perspective of POC TDM promises both real-time quantification and computer-assisted guidance for precision dosing, which will allow oncologists to perform all stages of drug level monitoring at the patient’s bedside without the need for costly measures in external laboratories. The mere concept of POC TDM raises multiple imperative concerns. Firstly, as previously mentioned, the sensitivity and selectivity of the device should match those of current standard analytical facilities. Secondly, the practical convenience of the device has to be optimized in order to accommodate both patients and health professionals at best. Additionally, the system should integrate an appropriate computer-assisted interpretation platform to assist the prescribers in dosage readjustment decisions. Lastly, both the measurement and the interpretation tailored to the patient’s characteristics must be reliable and properly validated, since the expertise of a pharmacologist may not be available on-site at all times. Currently evolving technologies for the precise and sensitive quantification of drugs [16] allow for considering with realism the possibility of a fruitful implementation of POC TDM in the relatively near future. Moreover, computer-assisted model-informed precision dosing software programs are sufficiently mature nowadays to offer model-based predictions and dosage adjustment suggestions to oncologists in the process of achieving the optimal target exposure for their patients. Information technologies will allow for the continuous improvement of the quality and adequacy of these tools. Together, these technologies could herald a new promising era for a more efficient use of conventional chemotherapies.

## Data Availability

No new data were created or analyzed in this study. Data sharing is not applicable to this article.

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
