# Peer review of "Precision Oncology by Point-of-Care Therapeutic Drug Monitoring and Dosage Adjustment of Conventional Cytotoxic Chemotherapies: A Perspective"

_pharmaceutics, 2023, doi:10.3390/pharmaceutics15041283_

Round 1

Reviewer 1 Report

The Authors reported the current limitation of the traditional therapeutic drug monitoring (TDM) of conventional cytotoxic chemotherapy agents. Six conventional antineoplastic drugs are examined. The following issues should be  addressed:

The abbreviation "PD" have to be clarified

 It is necessary to explain in more detail the PK-PD  approach

 Table 1 is “supplementary  materials”, however it should be linked to the text of the article with the insertion of more references/citations

Reviewer 2 Report

Improving efficacy, minimizing the adverse side effects of drugs, and overcoming acquired resistance to drug treatment have been major goals and emphases in cancer therapy. In order to attain this  objectives of precision oncology and   conventional cytotoxic chemotherapieis important to applay the  therapeutic drug monitoring . In the case of cytotoxic drugs they can be  very efficacious but  also have low therapeutic index.and so,  a great caution needs to be exercised in their usage. To optimize the efficacy these drugs need to be given at maximum tolerated dose which leads to significant amount of toxicity to the patient. The fine balance between efficacy and safety is the key to the success of cytotoxic chemotherapeutics. However, it is possibly more rewarding to obtain that balance for this class drugs as the frequency of drug related toxicities are higher compared to the other therapeutic class and are potentially life threatening and may cause prolonged morbidity. Significant efforts have been invested in last three to four decades in therapeutic drug monitoring (TDM) research to understand the relationship between the drug concentration and the response achieved for therapeutic efficacy as well as drug toxicity for cytotoxic drugs. TDM evolved over this period and the evidence gathered favored its routine use for certain drugs.  Since, TDM is an expensive endeavor both from economic and logistic point of view, to justify its use it is necessary to demonstrate that the implementation leads to perceivable improvement in the patient outcomes.Therefore, good quality data from well-designed observational study do add immense value to the scientific knowledge base, when they are examined in totality, despite the heterogeneity amongst them. This article aims to define the adjustment needed from current to efficient TDM practice for cytotoxics,  namely point-of-care (POC) TDM. For real-time dose adjustment, required for chemotherapies, such POC-TDM is only achievable with analytical methods that match the sensitivity and selectivity of current methods such as chromatography, as well as model-informed precision dosing platforms to assist the oncologist for dose fine-tuning based on quantification results and targeted intervals. t is very interesting

Reviewer 3 Report

Briki et al. presented examples and the significance of point-of-care therapeutic drug monitoring for cytotoxic chemotherapies in this manuscript. The study covers several aspects of the topic, including a brief introduction to current drugs and their potential applications and future perspectives. However, some suggestions for improvement can be made:

1. It is recommended that the authors include a table or timeline figure to illustrate the development of therapeutic drug monitoring for different chemotherapies.

2. The authors should provide information on the clinical outcomes of TDM-guided chemotherapy.

3. The manuscript should include a section that discusses the technical and logistical challenges associated with implementing TDM in clinical practice, as well as the cost-effectiveness of TDM-guided chemotherapy.

Author Response

Please the attachment
